

# Genetic polymorphisms and forensic efficiency of 19 X-chromosomal STR loci for Xinjiang Mongolian population

Ling Chen[1,*], Yuxin Guo[2,3,*], Cheng Xiao[1], Weibin Wu[1], Qiong Lan[1], Yating Fang[1], Jiangang Chen[4] and Bofeng Zhu[1,2,3]

[1] Department of Forensic Biology, School of Forensic Medicine, Southern Medical University, Guangzhou, Guangdong, China
[2] Key Laboratory of Shaanxi Province for Craniofacial Precision Medicine Research, College of Stomatology, Xi'an Jiaotong University, Xi'an, Shaanxi, China
[3] Clinical Research Center of Shaanxi Province for Dental and Maxillofacial Diseases, College of Stomatology, Xi'an Jiaotong University, Xi'an, Shaanxi, China
[4] Institute of Forensic Science, Ministry of Public Security, Beijing, China
* These authors contributed equally to this work.

Corresponding author
Bofeng Zhu,
zhubofeng@mail.xjtu.edu.cn

## ABSTRACT

**Aim:** X-chromosomal short tandem repeat (X-STR) loci are playing an increasingly important role in some complex kinship cases in recent years. To investigate the forensic efficiency of X-STRs of Mongolian minority group from Xinjiang Uygur Autonomous Region, China, and further depict the genetic relationship among Xinjiang Mongolians and other populations, 267 blood samples from unrelated healthy Xinjiang Mongolians were amplified by an AGCU X-19 STR kit.

**Results:** No deviations for all 19 X-STR loci were observed from the Hardy–Weinberg equilibrium after Bonferroni correction ($p > 0.0026$) in female samples. The most frequent allele was allele 10 at locus DXS10164 with the frequency 0.5663. The polymorphism information content values of the 19 X-STR loci were more than 0.5 with the highest polymorphism at the locus DXS10135. The cumulative power of discrimination were 0.9999999999999999999988761005481 in females and 0.999999999999903 in males, respectively; and the cumulative mean exclusion chances were 0.9999999969738068321121 in duos and 0.999999999998952 in trios, respectively. The seven linkage groups were extremely informative, with all the haplotype diversities greater than 0.9487. No linkage disequilibrium was observed for a significance level of 0.00029 ($p = 0.05/171$) after Bonferroni correction. The $D_A$ distances, multidimensional scaling plot and phylogenetic tree based on the 11 overlapping X-STR loci all presented that the Xinjiang Mongolian population was genetically different from other Asian populations, including the Mongolian population from Inner Mongolia Autonomous Region, China.

**Conclusion:** This study indicated that the 19 X-STR multiplex PCR system was of high utility value for both forensic practices and population genetic research in Xinjiang Mongolian group.

## INTRODUCTION

As one of the most famous nomadic nations in the world, the Mongols mainly live in East-Central Asian. With a population of 5.8 million people, Mongolian was the eighth largest ethnic minority on Chinese population data according to the sixth China population census in 2010 (http://www.stats.gov.cn/tjsj/pcsj/rkpc/6rp/indexch.htm). Their indigenous dialects are known as the Mongolian language, which belongs to Altaic language family. The traditional Mongolian script was created in the early 13th century based on the script of Huihu (*Janhunen, 2011*). Based on Chinese historical texts, the ancestry of Mongolians can be traced back to Donghu, a nomadic confederation of tribes with same ethnic origin, different dialects and names. After the Donghu were defeated by Xiongnu, the Xianbei and Wuhuan survived as the main remnants of the confederation. As recorded in the Chinese history, Xianbei split into three prominent groups: the Rouran, the Khitan and the Shiwei. A subtribe of Shiwei, called "Shiwei Menggu," was held to be the origin of Mogolians. In the 13th century, Genghis Khan united a large group of Mongolic-speaking tribes and founded the Mongol Empire. With the expansion of the Mongol Empire, the Mongolians settled over almost all Eurasia and carried on military campaigns from the Adriatic Sea to Indonesian Java island and from Japan to Palestine. In the late 14th century, Mongolia was divided into two parts: Western Mongolia (Oirats) and Eastern Mongolia (Khalkha, Inner Mongols, Barga, Buryats) after the fall of the Mongol Empire. In the modern period, the most prominent Mongol groups are the Inner Mongols concentrated in Inner Mongolia Autonomous Region and Northeast China, the Oirats concentrated in Xinjiang Uyghur Autonomous Region of China, and the Khalkha, known as Outer Mongols, concentrated in State of Mongolia. Approximately sixty percent Mongolians in the world settled in China (https://en.wikipedia.org/wiki/Mongols).

In recent years, autosomal short tandem repeats (STRs) have been extensively applied in paternity testing and individual identification. However, autosomal STRs may not be effective in some deficiency paternity cases, such as tests of a half-sister without the father's DNA, grandmother–granddaughter without the parent's DNA, paternal aunt–niece or maternal aunt/maternal uncle–nephew without the parent's and grandparent's DNA. X-chromosomal short tandem repeats (X-STRs) have special characteristics that make them particularly useful in the forensic cases described above (*Chen et al., 2014*; *Szibor, 2007*; *Szibor et al., 2003b*). Therefore, X-STR loci have drawn more and more attention in the forensic sciences and have a special role in forensic cases (*Liu et al., 2013*; *Sun et al., 2013*). At present, X-chromosomal genetic information of Mongolians had been reported by Hou et al. using nine X-STR Loci, Zhang et al. using 34 X-markers (18-STRs and 16-Indels), and Tao et al. using 12 X-STR loci (*Hou, Yu & Li, 2007*; *Tao et al., 2018*; *Zhang et al., 2015*). However, this research only involved in Inner Mongolians, and few studies are aimed at Western Mongols. The AGCU-X19 STR kit was a recently developed multiplex amplification system, which can amplify 19 X-STR loci simultaneously, including DXS8378, DXS7423, DXS10148, DXS10159, DXS6809, DXS7424, DXS10164, DXS10162, DXS7132, DXS10079, DXS6789, DXS101, DXS10103, DXS10101, HPRTB, DXS10075, DXS10074, DXS10135 and DXS10134. *Yang et al. (2016)* had indicated

it could be used as a supplementary tool in kinship tests in China. Furthermore, genetic polymorphisms of this X-STR system have been investigated in a number of populations (*Liu et al., 2017*; *Meng et al., 2017*; *Yang et al., 2017*; *Zhang et al., 2016*). In the present study, we used this 19 X-STR multiplex PCR system to obtain genetic information from Western Mongols, including the allele frequencies and forensic parameters of these 19 X-STR loci, the haplotypic diversities of seven X-STR linkage groups. We also estimated the population differentiations between the Western Mongols and other previously reported groups.

## MATERIALS AND METHODS

### Population samples

Blood samples were collected from 267 unrelated healthy Xinjiang Mongolians (156 males and 111 females) living in Bortala Mongol Autonomous Prefecture, Xinjiang Uyghur Autonomous Region, China. Genomic DNA was extracted using the Chelex-100 method (*Walsh, Metzger & Higuchi, 1991*).

### Compliance with ethics guidelines

This study was carried according to the approval for research involving human and animals by the ethical committee of Xi'an Jiaotong University Health Science Center, China. Informed consent was signed by all participants prior to sample collection.

### PCR amplification and STR typing

DNA samples were amplified in a 25 $\mu$l PCR reaction volume using the AGCU-X19 STR kit (AGCU ScienTech Incorporation, Wuxi, Jiangsu, China), following the manufacturer's instructions. The PCR products were separated through capillary electrophoresis with an ABI 3500xl Genetic Analyzer (Thermo Fisher Scientific, Waltham, MA, USA) and analyzed by GeneMapper ID-X (Thermo Fisher Scientific, Waltham, MA, USA).

### Quality control

This study was carried out strictly following the ISFG recommendations on the analysis of DNA polymorphisms (*Schneider, 2007*) and guidelines for publication of population data (*Carracedo et al., 2013*, *2014*). The experiments were conducted under the laboratory internal control standards. Negative control (deionized water for amplified reaction) and positive control (the 9947A DNA sample; Promega, Madison, WI, USA) were genotyped along with each batch of samples.

### Statistical analyses

Allelic frequencies of all the 19 X-STR loci were calculated using the PowerStats v1.2 program (Promega, Madison, WI, USA). Hardy–Weinberg equilibrium and pair-wise linkage disequilibrium in female samples were conducted by the software of Genepop (http://genepop.curtin.edu.au/). Differences of allele frequencies in males and females were assessed by the standard analysis of variance (ANOVA) method using statistical software SPSS Version 19.0 (IBM, SPSS Inc., Chicago, IL, USA). The haplotype frequencies in male samples were also calculated by SPSS Version 19.0. Polymorphism

information content (PIC), Heterozygosity (Het), power of exclusion (PE), power of discrimination in females (PDF), power of discrimination in males (PDM), mean exclusion chance (MEC) for deficiency cases, normal trios and duo cases were calculated using the free online-calculation tool of ChrX-STR.org 2.0 database (http://www.chrx-str.org). The geographical distributions of the 15 Asian populations were drew by the R software (https://www.r-project.org/). The $D_A$ distance and the phylogenetic tree were constructed using Poptree2 Software (*Takezaki, Nei & Tamura, 2010*). Based on the $D_A$ distances, the multidimensional scaling (MDS) plot was constructed by SPSS Version 19.0.

## RESULTS

### Allelic frequencies of 19 X-STR loci

A total of 156 male and 111 female individuals from Xinjiang Mongolian group were analyzed. And the raw genotypes at 19 X-STR loci are displayed in Table S1. Exact tests of HWE after Bonferroni correction ($p > 0.05/19 = 0.0026$) demonstrated no significant deviations for all 19 X-STR loci in female samples. There were no significant differences in the allele frequency distributions between males and females by the results of One-Way ANOVA ($p > 0.05/19 = 0.0026$), which indicated that the allele frequency distributions of all the 19 X-STR loci had no gender bias. Therefore, the allele frequencies of males and females were pooled together according to the formula (2*femalefreq + malefreq)/3, which are shown in Table S2. A total of 223 alleles were observed at the 19 X-STR loci, and the allele numbers ranged from 5 at DXS7423 to 26 at DXS10148. The most frequent allele observed was allele 10 at locus DXS10164 with the frequency 0.5663. Since all positive controls were genotyped correctly, we confirmed a series of off-ladder alleles for the 267 Mongolian individuals, among which allele 18.1 and 19.1 at DXS10079 and DXS10162 were more than 0.1.

### Forensic parameters of 19 X-STR loci in Xinjiang Mongolian group

Forensic statistical parameters of the 19 X-STR loci were calculated based on the pooled allele frequencies. As displayed in Table 1, all the 19 X-STR loci were found to be very informative (PIC > 0.5). DXS10135, with the highest PIC (0.9184), Het (0.9235), PE (0.8437), PDF (0.9890) and PDM (0.9235), had the most polymorphism. The lowest PIC (0.5334), Het (0.5915), PE (0.2808), PDF (0.7750) and PDM (0.5915) were observed at DXS10164, indicating that DXS10164 was of the lowest polymorphism. The cumulative PDF and PDM were 0.9999999999999999999988761005481 and 0.99999999999903, respectively. The cumulative PDF and PDM both showed extremely high values, which confirmed that the 19 X-STR multiplex system would be powerful for individual identification in the Xinjiang Mongolian group. The MEC Krüger, MEC Kishida, and MEC Desmarais Duo ranged from 0.3406 to 0.8448 for the deficiency cases, 0.5334–0.9183 for the normal trios and 0.3867–0.8539 for the duo cases, respectively. The cumulative MECs were 0.99999983245493 in deficiency cases, 0.999999999998952 in normal trios, and 0.9999999969738068321121 in duo cases, respectively. The high values of these cumulative MECs suggested that the 19 X-STR multiplex system could be useful in paternity tests, especially for some deficiency cases.

**Table 1 The forensic efficiency parameters of 19 X-STR loci in Xinjiang Mongolians.**

| Index | PIC | Het | PE | PDF | PDM | MEC Krüger | MEC Kishida | MEC Desmarais Duo |
|-------|-----|-----|-----|-----|-----|------------|-------------|-------------------|
| DXS8378 | 0.6157 | 0.6687 | 0.3815 | 0.8372 | 0.6687 | 0.4160 | 0.6157 | 0.4687 |
| DXS7423 | 0.5430 | 0.6077 | 0.3003 | 0.7814 | 0.6077 | 0.3442 | 0.5430 | 0.3966 |
| DXS10148 | 0.8980 | 0.9056 | 0.8069 | 0.9835 | 0.9056 | 0.8095 | 0.8980 | 0.8221 |
| DXS10159 | 0.7550 | 0.7882 | 0.5773 | 0.9220 | 0.7882 | 0.5817 | 0.7550 | 0.6254 |
| DXS10134 | 0.8494 | 0.8635 | 0.7217 | 0.9672 | 0.8635 | 0.7296 | 0.8494 | 0.7508 |
| DXS7424 | 0.7185 | 0.7499 | 0.5096 | 0.9061 | 0.7499 | 0.5457 | 0.7185 | 0.5830 |
| DXS10164 | 0.5334 | 0.5915 | 0.2808 | 0.7750 | 0.5915 | 0.3406 | 0.5334 | 0.3867 |
| DXS10162 | 0.6804 | 0.7199 | 0.4598 | 0.8820 | 0.7199 | 0.4958 | 0.6804 | 0.5393 |
| DXS7132 | 0.7047 | 0.7453 | 0.5018 | 0.8945 | 0.7453 | 0.5200 | 0.7047 | 0.5669 |
| DXS10079 | 0.7883 | 0.8130 | 0.6234 | 0.9403 | 0.8130 | 0.6345 | 0.7883 | 0.6681 |
| DXS6789 | 0.7679 | 0.7966 | 0.5927 | 0.9299 | 0.7966 | 0.6050 | 0.7679 | 0.6425 |
| DXS101 | 0.8119 | 0.8321 | 0.6600 | 0.9516 | 0.8321 | 0.6707 | 0.8119 | 0.6992 |
| DXS10103 | 0.7559 | 0.7863 | 0.5739 | 0.9240 | 0.7863 | 0.5876 | 0.7558 | 0.6271 |
| DXS10101 | 0.8931 | 0.9015 | 0.7984 | 0.9819 | 0.9015 | 0.8001 | 0.8930 | 0.8141 |
| HPRTB | 0.6781 | 0.7244 | 0.4669 | 0.8777 | 0.7244 | 0.4844 | 0.6781 | 0.5362 |
| DXS6809 | 0.7799 | 0.8051 | 0.6085 | 0.9368 | 0.8051 | 0.6232 | 0.7799 | 0.6572 |
| DXS10075 | 0.6289 | 0.6845 | 0.4047 | 0.8448 | 0.6845 | 0.4296 | 0.6289 | 0.4839 |
| DXS10074 | 0.7535 | 0.7847 | 0.5710 | 0.9225 | 0.7847 | 0.5845 | 0.7535 | 0.6244 |
| DXS10135 | 0.9184 | 0.9235 | 0.8437 | 0.9890 | 0.9235 | 0.8448 | 0.9183 | 0.8539 |

Note:
PIC, polymorphism information content; Het, heterozygosity; PE, probability of exclusion; PDF, power of discrimination in females; PDM, power of discrimination in males; MEC Krüger, mean exclusion chance for deficiency cases; MEC Kishida, mean exclusion chance for normal trios; MEC Desmarais Duo, mean exclusion chance in duo cases.

## Haplotypic structure of seven linkage groups

According to the study of *Zhang et al. (2016)*, the 19 X-STR loci could be divided into seven linkage groups. DXS8378-DXS10148-DXS10135, DXS7132-DXS10079-DXS10075-DXS10074, DXS10103-DXS10101-HPRTB and DXS10134-DXS7423 (http://xdb.qualitype.de/xdb/linkageTable.jsf) were classified as linkage groups 1, 2, 3 and 7, respectively. DXS6809-DXS6789 (*Pasino et al., 2011*; *Szibor et al., 2003a*) was specified as linkage group 4; DXS10159-DXS10162-DXS10164 (*Meng et al., 2014*; *Ye et al., 2014*) as linkage group 5; and DXS7424-DXS101 (*Edelmann et al., 2002*; *Szibor et al., 2003a*) as linkage group 6. The haplotypic frequencies and haplotypic diversities of the seven linkage groups are shown in Table S3. In this study, 127 haplotypes were observed for linkage group 1, 117 haplotypes for linkage group 2, 84 haplotypes for linkage group 3, 43 haplotypes for linkage group 4, 53 haplotypes for linkage group 5, 41 haplotypes for linkage group 6, 34 haplotypes for linkage group 7. The seven linkage groups were proved to be fairly informative in the Xinjiang Mongolians because of the high values of haplotype diversities, which ranged from 0.9487 at linkage group 7 to 0.9969 at linkage group 1. Among these linkage groups, the most common haplotypes were H 33-20 for linkage group DXS6809-DXS6789 and H 16-24 for linkage group DXS7424-DXS101 with

a frequency of 0.1090, followed by H 37-15 for linkage group DXS10134-DXS7423 with a frequency of 0.1026. For linkage group DXS8378-DXS10148-DXS10135, which had a large variety of haplotypes with relatively low frequencies, the most common haplotype was H 10-25.1-21, with a frequency of 0.0256. However, H 10-25.1-21 was not found in the Xibe population (*Meng et al., 2017*) and was observed in the Guanzhong Han population (*Zhang et al., 2016*) with a relatively lower frequency of 0.0040. For linkage group DXS10103-DXS10101-HPRTB, the most common haplotype in the Xinjiang Mongolians was H 19-30.2-12 with a frequency of 0.0449, followed by H 16-32-13 with 0.0385; while in the Japanese population (*Uchigasaki, Tie & Takahashi, 2013*), H 19-30.2-12 and H 16-32-13 were at a frequency of 0.0091, H 16-30-13 with a frequency of 0.0411 was the most common haplotype instead. For linkage group DXS10159-DXS10162-DXS10164, H 24-18-10 had the highest frequency of 0.0833; however, H 26-18-10 was much more common than H 24-18-10 in the Xibe (*Meng et al., 2017*) and Guangxi Han populations (*Ye et al., 2014*). For the linkage group DXS7424-DXS101, H 16-24 was of a medium frequency of 0.0403 in the Kazak population (*Liu et al., 2017*), where H 16-25 with a frequency of 0.0872 was the most common haplotype. These haplotype comparisons indicated that the studied Xinjiang Mongolian population had a different pattern of haplotype distribution, compared to other Asian populations.

## Linkage disequilibrium analyses

Pair-wise linkage disequilibrium in the female samples were analyzed by Genepop software before the 19 X-STR loci were applied for forensic purposes and population genetics. In this study, 171 pair-wise comparisons were performed (Table S4) and the results revealed that 10 pairs of pair-wise loci had $p$-value < 0.05. Nevertheless, no linkage disequilibrium was observed for a significance level of 0.00029 ($p = 0.05/171$) after Bonferroni correction. This indicated that there were not non-random association of alleles at different loci in the Xinjiang Mongolian group.

## Inter-population comparisons based on the 11 overlapping X-STRs

To investigate the population differentiations between the studied Xinjiang Mongolians and other Asian groups, we collected allele frequencies of 11 overlapping X-STR loci (DXS8378, DXS7423, DXS10148, DXS10134, DXS7132, DXS10079, DXS10103, DXS10101, DXS10074, DXS10135 and HPRTB) from Xibe (*Meng et al., 2017*), Kazak (*Liu et al., 2017*), Inner Mongolian (*Tao et al., 2018*), Japanese (*Uchigasaki, Tie & Takahashi, 2013*), Shanghai Han (*Zhang et al., 2012*), Bhil (*Shrivastava et al., 2015*), Malay (*Samejima et al., 2012*), Taiwanese (*Chen et al., 2014*), Ili Uygur (*Guo et al., 2016*), Guanzhong Han (*Zhang et al., 2016*), Tibetan (*Yang et al., 2017*), Southern Han (*Yang et al., 2017*), Hui (*Yang et al., 2017*) and Korla Uygur (*Yang et al., 2017*). For a better understanding, the geographical distributions of these populations mentioned above are shown in Fig. 1. The $D_A$ distance (*Nei, Tajima & Tateno, 1983*) was often used in forensic science for measuring genetic distance between populations, and the $D_A$ distance was more efficient in obtaining the correct tree topology for microsatellite data than other distance measures including $D_{ST}$ and $F_{ST}$ (*Takezaki & Nei, 1996, 2008*).

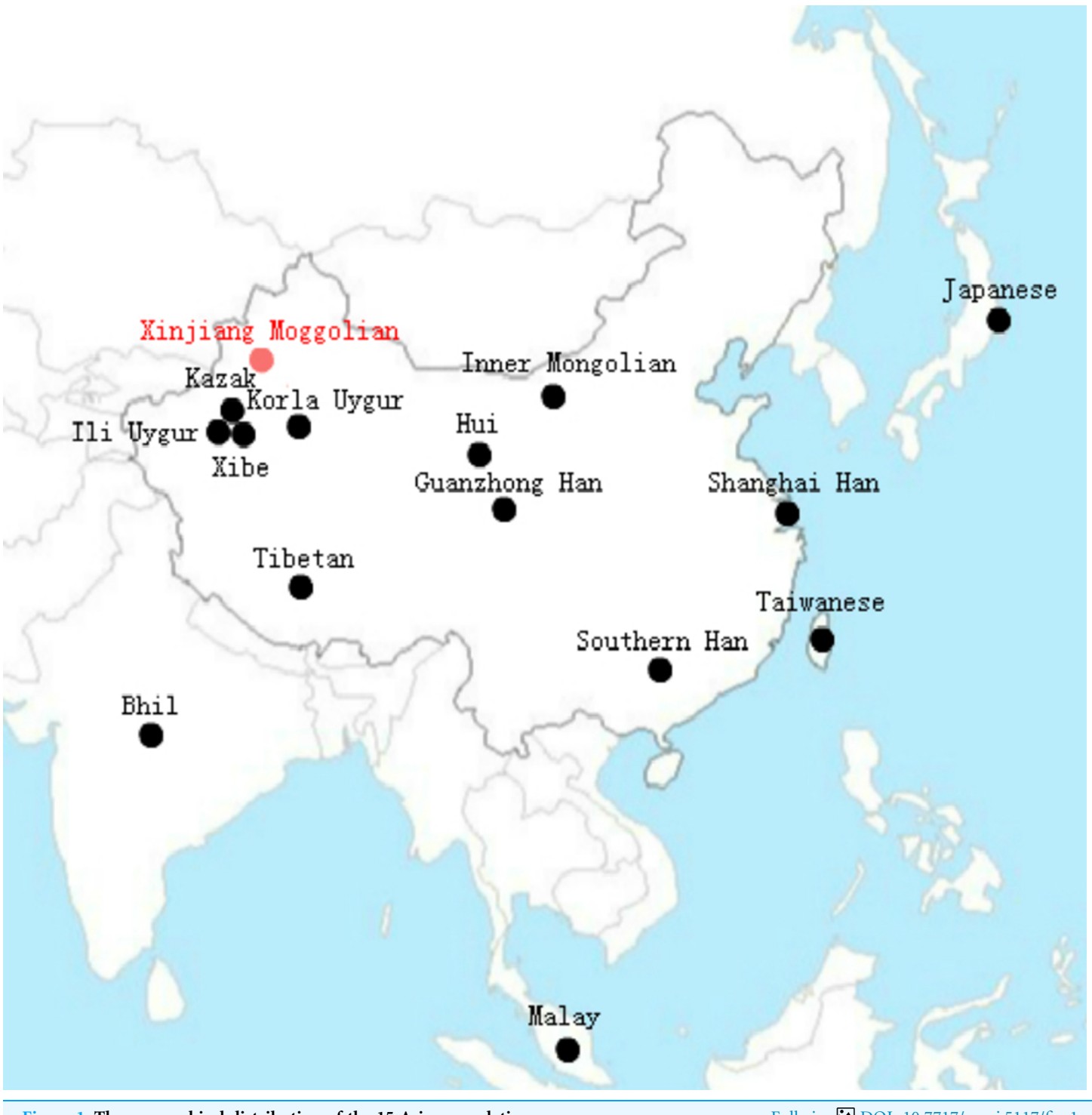

**Figure 1 The geographical distribution of the 15 Asian populations.**

Therefore, through the comparisons of allelic frequencies of the 11 overlapping X-STR loci, we calculated the $D_A$ distances between these Asian populations. As shown in Table S5, the largest $D_A$ distance was observed between Xinjiang Mongolian and Bhil

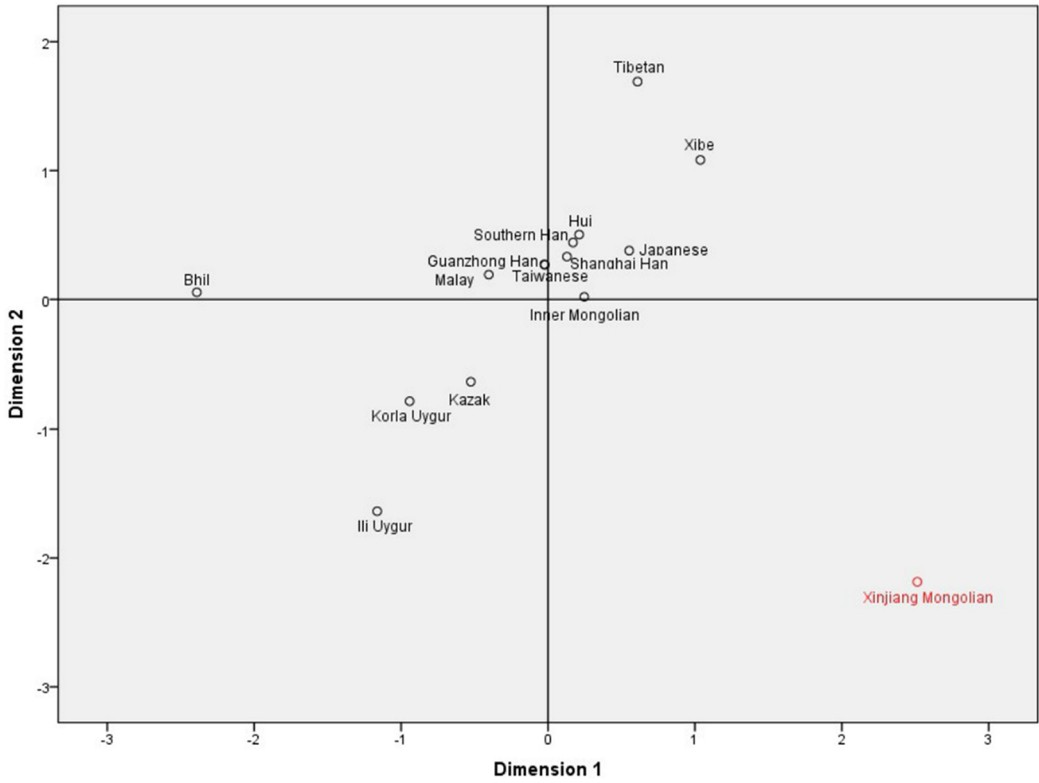

**Figure 2 The multidimensional scaling (MDS) plot based on $D_A$ distances among the 15 Asian populations.**

(0.078), and the smallest $D_A$ distance between Guanzhong Han and Taiwanese (0.009). Based on these $D_A$ distances, a MDS plot was constructed. As shown in Fig. 2, the Southern Han, Guanzhong Han, Shanghai Han, Taiwanese, Japanese, Hui, Malay, as well as Inner Mongolian were clustered together at the center part of this plot; however, only the Xinjiang Mongolian was located in the lower right corner, keeping a long distance from other populations. Thus, we can see that the Xinjiang Mongolian population was obviously different from other Asian populations, including Inner Mongolian.

## Phylogenetic analyses based on the 11 overlapping X-STRs

To evaluate the evolutionary relationship between the Xinjiang Mongolian and other 14 Asian populations, we constructed a phylogenetic tree using the $D_A$ distances and the unweighted pair-group method with arithmetic means method. As shown in Fig. 3, populations with the same ethno-linguistic origin such as Taiwanese, Guanzhong Han, Shanghai Han, Southern Han, Japanese were clustered together; populations with close geographic distance like Korla Uygur, Kazak were clustered as a terminal clade; only the Xinjiang Mongolian was segregated as a distant outlier, indicating that the Xinjiang Mongolian population was genetically distinct from those compared Asian populations. On the whole, the phylogenetic tree showed a genetic relationship similar to the result of MDS plot.

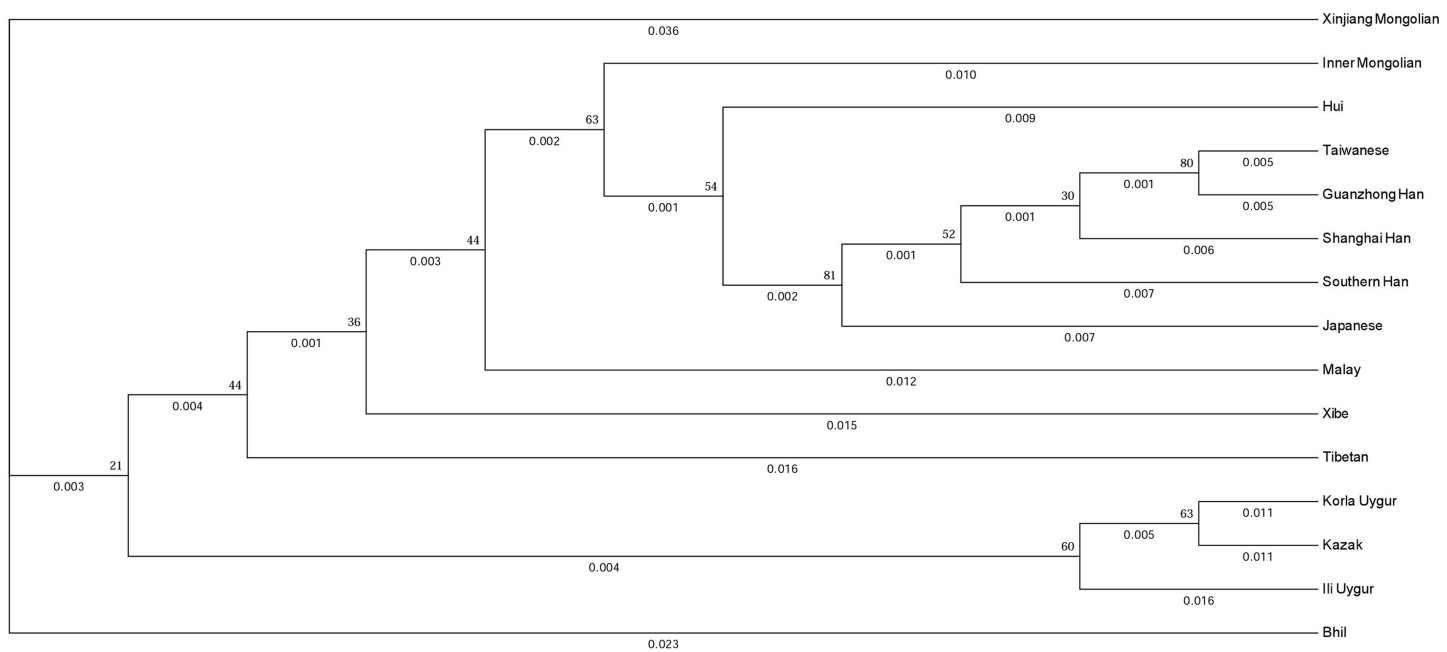

**Figure 3** The phylogenetic tree generated by the $D_A$ distance and the unweighted pair-group method with arithmetic means method (UPGMA).

## DISCUSSION

Results of this study showed that the 19 X-STR multiplex PCR system was very useful for forensic applications and population genetic researches in Xinjiang Mongolian population. Compared with Investigator Argus X-12 amplification kit (Qiagen, Hilden, Germany), the AGCU 19 X-STR multiplex PCR system enriched eight new X-STR loci (DXS10159, DXS10162, DXS7424, DXS10164, DXS6789, DXS101, DXS6809 and DXS10075). The 19 X-STR multiplex PCR system has been studied in several Chinese populations such as Zhejiang Han (*Yang et al., 2016*), Guanzhong Han (*Zhang et al., 2016*), Kazak (*Liu et al., 2017*), Xibe (*Meng et al., 2017*), Uygur (*Guo et al., 2016*), and these studies also indicated its potential in paternity testing and individual identification. However, there has been no data of the set of 19 X-STR for Western Mongolians so far. There were about 170,000 Mongolians residing in Xinjiang Uyghur Autonomous Region, China according to the sixth China population census in 2010 (http://www.stats.gov.cn/tjsj/pcsj/rkpc/6rp/indexch.htm). Thus, we performed a study on the polymorphism of 19 X-STRs in Xinjiang Mongolians. In summary, the forensic statistical parameters of the 19 X-STR loci for Xinjiang Mongolian population showed that most loci owned high PIC, Het and PE, and the new added eight X-STRs were highly polymorphic except DXS10164 (PIC = 0.5334). The combined power of discrimination was more than 0.999999999999 for both the female samples and the male samples. Moreover, the combined MEC was more than 0.99999999999 in normal trio cases and over 0.99999999 in duo cases. These results suggested that the 19 X-STR multiplex system could provide highly polymorphic information suitable for individual identification and paternity

testing in Xinjiang Mongolian populations. Especially, the combined MEC was more than 0.9999999 in deficiency cases. This meant that the 19 X-STRs could be helpful for deficiency paternity cases.

As mentioned above, the 19 X-STR loci were grouped into seven linkage groups. The results showed that seven linkage groups could provide highly haplotype diversity information for the Xinjiang Mongolian population. The obtained X-chromosomal haplotype frequencies of the seven linkage groups could be essential for calculating likelihood ratio of kinship. Previous research showed that the haplotyping of the X-STR loci could be used to analyze some complex kinship testing (*Edelmann et al., 2002*; *Szibor, 2007*). The MDS plot and the phylogenetic tree based on the 11 overlapping X-STRs all presented that the Xinjiang Mongolian population was genetically different from other Asian populations, including the Mongolians from Inner Mongolia Autonomous Region, China.

The Xinjiang Mongolians were the subgroup of the Oirats, whose ancestral home was in the Altai region of western Mongolia. The Oirats share some history, geography, culture and language with the Eastern Mongols. However, they were often at war with the Eastern Mongols, only few times united as a larger Mongol entity. Gradually, the Oirats and Eastern Mongols had developed separate identities after the collapse of the Mongol Empire (https://en.wikipedia.org/wiki/Mongols).

Previous studies based on X-chromosomal STR loci indicated that Mongolians were closely related with Chinese Han groups (*Hou, Yu & Li, 2007*; *Tao et al., 2018*; *Zhang et al., 2015*). In fact, the Mongolians involved in these studies were only Inner Mongolians, descendants of the Eastern Mongols. Through the analysis of autosomal SNPs, *Xing et al. (2013)* indicated that Deedu Mongolians, living in the Qinghai-Tibetan Plateau, shared genetic ancestry with other Mongolians as well as Tibetan populations; *Nakayama et al. (2017)* demonstrated that the largest fraction of the ancestry of Outer Mongols was mainly shared by ethnic groups in Central and Northeast China (Daur, Hezhen, Oroqen, and Xibe), and by Siberians (Yakut), and this fraction was present to a lesser extent in Southern populations including Japanese, Chinese Han, and other ethnic groups in Southeast Asia. Thus, we can see that many factors like the sample sizes, the origins of the selected samples, the choice of the genetic markers and the coverage of reference populations may lead to the discrepancy in the analysis of genetic relationships. In summary, to achieve a better understanding of genetic relationships between those populations, larger sample sizes, more genetic markers, more population data and further investigations are needed.

## CONCLUSION

In this study, 267 unrelated individuals from the Chinese Mongolian minority living in Xinjiang Uyghur Autonomous Region of China were genotyped and analyzed for 19 X-STR loci for the first time. These loci were proved to be highly polymorphic and demonstrated high power of discrimination in the Xinjiang Mongolian minority. The present study indicated that the 19 X-STR multiplex PCR system could be of high utility value for forensic practices, especially for the deficient paternity cases. We also found that

the Xinjiang Mongolian population was genetically different from other Asian populations, including the Mongolian population from Inner Mongolia Autonomous Region, China.

## ACKNOWLEDGEMENTS

The authors would like to thank Jiangwei Yan for helping us to analysis data.

### Funding
This project was supported by the National Natural Science Foundation of China (NSFC, No. 81772031), GDUPS (2017). The funders had no role in study design, data collection and analysis, decision to publish, or preparation of the manuscript.

### Grant Disclosures
The following grant information was disclosed by the authors:
National Natural Science Foundation of China (NSFC): 81772031.

### Competing Interests
The authors declare that they have no competing interests.

### Author Contributions
- Ling Chen conceived and designed the experiments, authored or reviewed drafts of the paper, approved the final draft.
- Yuxin Guo analyzed the data, prepared figures and/or tables, authored or reviewed drafts of the paper, approved the final draft.
- Cheng Xiao performed the experiments, prepared figures and/or tables, approved the final draft.
- Weibin Wu performed the experiments.
- Qiong Lan performed the experiments.
- Yating Fang performed the experiments.
- Jiangang Chen contributed reagents/materials/analysis tools.
- Bofeng Zhu conceived and designed the experiments, authored or reviewed drafts of the paper.

### Human Ethics
The following information was supplied relating to ethical approvals (i.e., approving body and any reference numbers):

This study was carried out according to the humane and ethical research principles approved by the ethical committee of Xi'an Jiaotong University Health Science Center, China.

### Data Availability
The raw data are provided in the Supplemental Files.

## Supplemental Information

Supplemental information for this article can be found online at http://dx.doi.org/10.7717/peerj.5117#supplemental-information.

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
