# Peer review of "Genetic polymorphisms and forensic efficiency of 19 X-chromosomal STR loci for Xinjiang Mongolian population"

_PeerJ, doi:10.7717/peerj.5117_

## Round 0.1 · original submission · Major Revisions

This is an interesting paper. However, the paper hasn't been sent for peer review yet, as I wanted to ask for some revisions before doing so.

Given that the authors have published another paper on a similar topic in Scientific Reports ("Genetic analysis of 19 X ..."), I would wish to have a combined analysis of data previously reported with the new data reported in this paper (i.e., include those from Uygurs, Hans, Tibetans, etc.). That would substantially improve the quality with a broader perspective. In fact, some data from the previous study seems to have already included in some of the analyses. Why not include them all?

I also ask the authors to provide the following:

1. As this paper, as well as the previous one in Scientific Reports, is aiming to understand human variation in time and space. It is crucial to include coordinates (latitude and longitude) of the samples. Please include such geographic information not only for the 267 samples in the raw data in your Supplemental Table 2, but also for those samples reported in that Scientific Reports paper included in the combined analysis. Such data will not only help readers to appreciate the spatial scale, but also attract more attention from scientists and increase visibility of the paper.

2. The raw data table in this paper (Supplemental Table 2) does not specify which is male and which female. Please include such information (in addition to the latitude and longitude that I mentioned in #1).

The references for multi-author papers were incorrectly formatted. Please reformat to increase readability.

If you and your co-authors could do the above, I am fairly certain that the revision will get positive reviews when I send it out for review.

I look forward to reading a revised manuscript. Thank you for sharing your interesting study with us.

---

## Round 0.2 · Major Revisions

I apologize for the delay in the reviewing process, but it is worth the wait. The three reviewers provided excellent and detailed comments on how to revise the manuscript. I am sure that you will appreciate their effort as much as I do, and I look forward to receiving an improved submission. Thank you again for your submission to PeerJ.

Reviewer 1 ·

Basic reporting

The authors should mention about several previous studies dealing autosomal SNPs in Mongolians populations (for instance, Xing et al. PLOS Genet. 2013, 9:e1003634, Nakayama et al. Mol Biol Evol 2017, 34:1936-1946, and more). Because autosomal SNPs are now common as targets for studying population diversity.

Experimental design

No comment

Validity of the findings

No comment

Additional comments

Aim of Abstract does not seem to properly show the purpose of this study. The should authors re-write to emphasize that testing the forensic efficiency of X chromosome STRs was their first interest, or include more precise population genetic analyses to depict genetic diversity of X linked loci in Mongolian populations, especially focusing on comparison with populations.

The authors should provide deeper discussion about diversity of X chromosome STRs. Comparison of Mongolians and other Asian populations in haplotypic diversity may be useful to give insights into the demographic history of these populations and it would be of interest to readers with biological anthropology background.

Reviewer 2 ·

Basic reporting

This paper presents the analyses of the 19 X-STR loci included in the AGCU-X19 STR kit in 267 Mongolian individuals from the Xinjiang province. Allele frequencies, Hardy-Weinberg equilibrium and parameters of forensic interest were estimated and a comparison with data from the literature was conducted. The present work enlarges the Chinese ethnic groups analyzed for this X-STR set in Yang et al. (2016), studying an important minority Chinese group.

Raw data. I thank you for providing the raw data, however I cannot understand the data in Supplementary Table 1. Authors supply the results of the 19 X-STRs in 267 individuals (men and women), but why women have only one allele in each STR? It is absolutely incomprehensible for me!

In general, along the text one can find a lot of mistakes (for instance line 175 DXS10134); expressions like “quite high values or “quite useful” that are very vague; redundant words (Mongolian groups lines 42-45) or sentences (lines 36-38, lines 121-123); not clear information (lines 96-97 “it was worth mentioning that….why it is worth mentioning?); grammatical errors (“these authors are contributed…”, line 60 “humane”, line 137 “wasn’t”)...The language and the structure should be improved.

Figure 2 is not clear or informative, it can be eliminated.
Table 3 should be checked.

Experimental design

The present study has been done following correct ethical and technical standards and lab methods are described with sufficient detail. Although authors should provide more justification about the Quality Control. ISFG guidelines (Carracedo et al. 2014) only indicate the number of samples to be analyzed to publish in ISFG Genetics…it is not a Quality Control described by Parson.

Regarding statistical analyses, these should be improved upod before Acceptance (as I have indicated in the general comments).

Although the study contains a large enough number of samples from an interesting population and therefore is of forensic and population genetic interest, in my opinion the manuscript needs a major revision before being considered suitable for publication in PeerJ.

Validity of the findings

The matter of study is not completely novel. In the introduction authors point out that only a nine X-STRs have been studied in the Mongolian group. But, in fact Zhang et al. (2015) analyzed 34 X-markers (18-STRs and 16-Indels) in this ethnic group and recently Tao et al. (2018) have studied 12 out of the 19 X-STR in Mongolians. Anyway, to study this set of X-STRs is useful provided that other Chinese populations have been previously typed with the same markers.

Although your data are useful in forensic casework and interesting for Population Genetic research, the conclusion that Mongolians are away from other clusters and that they have a unique genetic structure is not supported by the results or by previous studies, where Mongolians showed to be closely related with other Chinese groups (Zhang et al. 2015, Tao et al. 2018 and others). These previous results should be included in the discussion. And, if the authors want to demostrate that with these 19 X-STRs Mongolians have an unique structure, additional analysis should be carried out, for instance STRUCTURE analysis.

In the General comments I have pointed out some points in the statistical methods and their interpretation that could be improved.

Additional comments

I would suggest some observations in order to improve the general content of the manuscript:
1-Tittle. The present tittle is not informative about the subject of the study, it should indicate the number of STRs or even the name of the kit, so readers can easily know which STRs are analyzed.

2-More information of the studied ethnic group should be provided. Origin, language, number of individuals, historical relationship with other Chinese ethnic groups…And, is the present sample representative of all the Mongolian population?

3-Statistical methods
a- Please, check PI values in Table 2. The Paternity Index values are not correct, since the more variable a marker is, the lower PI value is provided. Logically it should be the opposite.

b-My main concern is in the comparison with other populations and the conclusions that authors draw from it.
-Which has been the criterion of selection of the populations? I agree with the Asian samples, due to the geographical context of the study, but there are other worldwide studies besides Tomas et al. (2012). Probably Greenlanders, Danes and Somalis are not the most suitable populations to be compared with Mongolians.

-The authors have calculated significant differences (Fst) in each STR…this information can be useful to evaluate the power of the different markers in population structure analyses, but not for evaluate the relationship between populations. Comparisons between populations should be done with the complete set of X-STRs.

-NJ tree (Figure 3) has not bootstrap values; therefore the significance of the clusters cannot be assessed. The genetic distances between populations would be better represented as a multidimensional scaling plot than a NJ tree of populations as in the present Figure. Such plot is simpler to interpret.

-The conclusion that Mongolians are away from other clusters and that they have a unique genetic structure is not supported by the results or by previous studies, where Mongolians showed to be closely related with other Chinese groups (Zhang et al. 2015, Tao et al. 2018 and others). These previous results should be included in the discussion. And, if the authors want to demostrate that with these 19 X-STRs Mongolians have an unique structure, additional analysis should be carried out, for instance STRUCTURE analysis.

-I have received two different abstracts, in the first, authors claim “a close affinity between Mongolian and Chinese Han populations”, that probably is more certain that the sentence in the other abstract “Mongolian groups had a unique genetic structure, which was significatively different from other populations”.

In conclusion, I cannot recommend the manuscript to be published in PeerJ in the present form. I would recommend that the manuscript was carefully checked, improving the analyses, the discussion and the language, and resubmission, providing the points raised were addressed.

Reviewer 3 ·

Basic reporting

This study analyses the 19 X-STR markers included in the AGCU-X19 STR kit organised in 7 linkage groups. In this work, 267 individuals from the minority Mongolian group in China are genotyped. Allele frequencies, Hardy-Weinberg equilibrium and forensic parameters are calculated for intrapopulation analysis. Interpopulation comparison is also performed.
Along the text, the English language should be improved. There are grammatical and typo mistakes for examples in lines: 15 (are), 36 (and), 60 (humane), 127 (but), etc. There are also problems with the conjugation of verbs (line 85, were drew drawn). In general, there is a mix of present and past tenses that would be good to equal. Sometimes the phrasing is redundant and make is difficult to follow (lines 121-122, 128,131,etc.).
In the introduction, I have missed more information about the population, historical events and how this minority group has been in contact with the other groups in China.
The reference citing has some mistakes in the text (lines 87, 120, 125, 165, 193…) and in the references part (line 225, this reference has three authors, please revise it).
Tables and Figures:
Figure 1: There is a small square in the bottom right part of the figure that I cannot see what it is. Do the different green colours in the map have any meaning? If they have should be specified in the Figure legend, if not, I would recommend putting all the regions in white.
Figure 2: This figure is not relevant. Can be deleted.
Figure 3: This figure gives not clear information. Since it is based on genetic distances, I would recommend changing it for the genetic distances matrix, then the interpretation of the results will be much clearer to see.
Figure 4: There is a problem with the nomenclature of the populations. It would be good if the authors always use of the name of the country or the gentilic (name of the inhabitants) but not a mix of them (this also happen in lines 141-147 in the text). Besides, the inhabitants of Denmark are called Danish, not Dane.
Table 1: In this table would be useful to label those alleles which are described for the first time. There is also some “O” instead of zeros (Allele 29 of DXS101, allele 31 of DXS10135). The frequency of allele 31 of DXS6809 has a “/” instead of a dot.
Table 2: The expected heterozygosity is described as HE at the bottom and called He in the Table. The PI value has no sense. It should be higher as more polymorphic is a marker, revise.
Table 3: There is different number of decimals in the table, I would recommend having the same number for all the data in the table. The format (the columns are not wide enough for all the data) makes it difficult to read the table.
Supplementary Table 1: The table presents the raw data of supposedly 267 individuals (111 women and 156 men). The main problem is that there is just one row for sample. Considering that X-chromosome markers are studied, should be two alleles per woman for each marker, and just one for males. So, I can’t understand how these markers have been genotyped and what is worse, how all the analyses in this paper have been done.
Supplementary Table 2: In this table should be specified, which markers belong to each Linkage group (at least in the legend), since the naming of the linkage groups is different that the proposed in Argus X kit (Qiagen) and even different from the used in Yang et al. (2017).
Supplementary Table 3: The format of this table is different from the other (the decimals in the frequencies have commas instead of points).
As I have mentioned before, Raw data is supplied (In Supplementary table 1) but the data has no sense. For that reason, is difficult to be sure if the analyses done in this study are done with this Raw data, or there is a mistake just in the table. Please check it, because it could do none of the analyses trustable.

Experimental design

Even though is not the first study on X-STR in this population, this paper is the first using this specific kit. I have missed an explanation on which marker contain this kit, I only could find it in the Tables. The number of samples is adequate.
Apart of the issue of the Raw data, and so of the number of chromosome studied, It seem strange for me, why arlequin software is used for allele calculations, but other parameters as haplotype frequencies are made by hand.
Regarding allele frequencies the best way to pool male and female frequencies is using the formula ( (2*femalefreq + male freq) /3), since woman has 2/3 of the chromosomes of the population. I am not sure how authors have pooled male and female data in this study.
I also wonder if POPTREE calculations are made in the correct form. This software need the number of chromosomes studied and allele frequencies in every population to calculate the genetic distances. These genetic distances are much better to compare populations differentiation that the Fst distances marker by marker (Table 3).

Validity of the findings

Since the result are interesting mainly in forensic genetics, and in population genetics (however only eleven markers are used for that part) the discussion of the paper is poor. The discussion is mostly a repetition of the data presented in results (lines 106-122 and 184-188, for example). I would expect the comparison with other studies using this X-STR kit in other populations, and also results obtained with other set of markers in this Mongolian population.
Then the authors would have more argument to state whether the Mongolians are similar to other groups in China or if they are statistically different from all of them. Since the conclusions they state are not supported and well argumented with the data they present.

Additional comments

Apart from all the commented before there are some points that should be addressed:
Line 175: DXS10134 is present in the Argus kit, the one that the author have studied and is not in the Argus is DXS10162.
Line 39: I miss a reference here, it is not clear which is the previous study you are referring to.
Lines 49 and 56: The name of the region is Borxtala or Bortala?
Line 103: There is an extra “0.” In the Ho value.
Line 126-128: Do this population have any historical relation with the Somalis? If not, this sentence is not relevant.

---

## Round 0.3 · accepted · Accept

I have carefully gone the “track_changes...” document and found the responses reasonable.

You may benefit from geophylogenies which provide temporal and geographic visualization of biodiversity. Here are some illustrations of a geophylogeny:

http://dambe.bio.uottawa.ca/PGT/PGT_raised.PNG
http://dambe.bio.uottawa.ca/PGT/PGT_flat_regular.PNG
http://dambe.bio.uottawa.ca/PGT/PGT_flat_terrain.PNG

Producing such geophylogenies is extremely easy (just a few clicks in fact) by using software such as PGT at:

http://dambe.bio.uottawa.ca/PGT/PGT.aspx

You are recommended to use geophylogeny, but you don't have to.

#